# Factorized Neural Radiance Field with Depth Covariance Function for Dense RGB Mapping

## Abstract

Reconstructing high-quality and real-time dense maps is critical for building the 3D environment for robot sensing and navigation. Recently, Neural Radiance Field (NeRF) has garnered great attention due to its excellent scene representation capacity of the 3D world; therefore, recent works leverage NeRF to learn 3D maps, typically based on RGB-D cameras. However, depth sensors are not always available for all devices, while RGB cameras are cheap and widely applicable. Therefore, we propose to use single RGB input for the scene reconstruction with NeRF, which becomes highly challenging without geometric guidance from depth sensors. Moreover, we cultivate its real-time capability with lightweight implementation. In this paper, we propose **FMapping**, a factorized NeRF mapping framework, allowing for high-quality and real-time reconstruction with only the RGB input. The insight of our method is that depth doesn't experience much change in consecutive RGB frames, thus the geometrical clues can be derived from RGB effectively with well estimated depth priors. In detail, we divide the mapping into 1) the initialization stage and 2) the on-the-fly stage. First, given trackers are not always stable in the initialization stage, we start with a noisy pose input to optimize the mapping. To this end, we exploit geometric consistency between volume rendering and signed distance function in a self-supervised way to capture depth accurately. In the second stage, given relatively short optimization time for real-time performance, we model the depth estimation as a Gaussian process (GP) with a pre-trained cost-effective depth covariance function to promptly infer depth on the condition of previous frames. Meanwhile, the per-pixel depth estimation and its corresponding uncertainty can guide the NeRF sampling process. Hence, we propose to densely allocate sample points within adjustable truncation regions near the surface, and further distribute samples to ones with high uncertainty. This way, we can continue building maps from subsequent poses with stabilized trackers. Experiments demonstrate that our framework outperforms state-of-the-art RGB-based mapping and achieves comparable performance to RGB-D mapping in terms of photometric and geometric accuracy, with real-time depth estimation capability in around 5 $Hz$ with consistent scale.

## 1 Introduction

Robot sensing and navigation rely on building high-quality dense maps in real-time, which provides instant feedback on the environment. Such a paradigm offers notable advantages by providing a comprehensive and instant scene reconstruction, beneficial for onboard tasks, such as robot navigation (Temeltas & Kayak, 2008; Fang et al., 2021) and interactive digital applications (Bettens et al., 2020; Sato et al., 2020). Earlier methods, e. g. , Henry et al. (2014); Dai et al. (2017b) are built based on RGB-D cameras, while their explicitly-cached point clouds impose a high requirement for computation and memory, limiting the practical application to resource-restricted mobile devices.

Neural Radiance Fields (NeRFs) (Mildenhall et al., 2020) have recently emerged as a compelling solution to the mapping problem for 3D reconstruction. For example, Sucar et al. (2021) and Zhu et al. (2022) propose to build neural implicit representations. In other words, NeRF utilizes latent representations, such as a multi-layer perceptron (MLP) to implicitly infer density and color of 3D points instead of caching them directly, thereby reducing memory consumption. These methods are not applicable when no depth sensors are available, as they often have difficulty in estimating

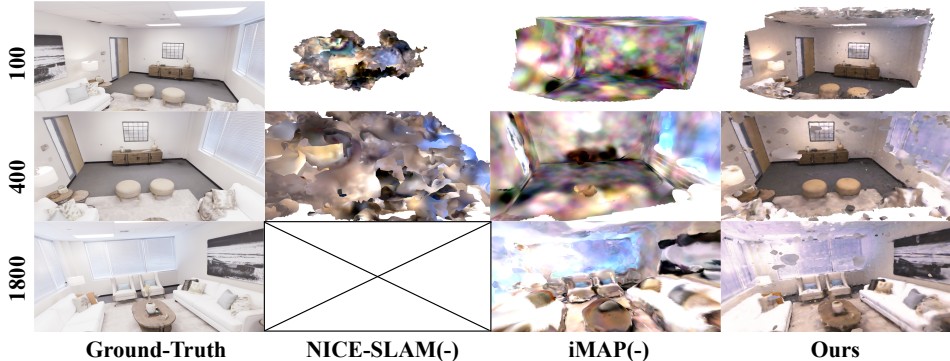

Figure 1: A simple experiment demonstrates the difficulty of real-time scene reconstruction by neural implicit representation without depth supervision. Dense mapping snapshots (at 100, 400, and 1800 input frames) of the on-the-fly running of NICE-SLAM(-) (Zhu et al., 2022), iMAP(-) (Sucar et al., 2021) and our method are displayed for Replica (Room0) sequence, given ground truth (GT) poses without depth supervision. (-) denotes that we make modifications to the original implementations by eliminating the back-propagation of the gradient from depth supervision.

accurate geometric cues. On the other hand, RGB cameras are widely applied with extensive usage, while lack of geometry guidance further poses a challenge on model's convergence during training. Therefore it is valuable to consider real-time mapping with pure RGB input. Some recent NeRF-based methods (Rosinol et al., 2022; Li et al., 2023; Zhu et al., 2023; Chung et al., 2022) have made similar efforts to tackle dense RGB mapping given more geometric constraints. For instance, Rosinol et al. (2022) incorporates geometric cues, e. g. , point clouds, derived from external SLAM systems (Teed & Deng, 2021). Li et al. (2023) performs multi-scale grid occupancy estimation using a cross-frame photometric warping loss. However, these methods require extensive computation to obtain extra geometrical evidence to reach a similar quality from the RGB-D SLAM system. Most of them can not be real-time without customized CUDA implementations.

Besides computation burden, mapping brings unique difficulties, which can be categorized into two primary aspects. (1) **Unstable trackers for pose initialization**: Facing the unknown environment, as observed by Cheng et al. (2021), trackers often do not perform well and have slow convergence speed. Therefore, it would be more realistic to learn mapping from the scratch without poses from trackers. (2) **Slow mapping within limited time**: During on-the-fly stage equipped with trackers, it requires a mapping system adaptable to growing scenes in real-time, while capable of rendering high-quality map. To our best knowledge, most of existing works scarify their real-time performance for high quality map reconstruction. Recently, $H_2$-Mapping (Jiang et al., 2023) attempts to solve the same issue by proposing a hybrid representation that combines octrees and implicit multi-resolution hash encoding to build maps with RGB-D cameras. However, its memory consumption is considerably large given such representation. On the contrary, we tackle to RGB dense mapping problem with lightweight Factorized NeRF, achieving comparable mapping quality.

In light of this, we present **FMapping**, an efficient neural field mapping framework that facilitates the continuous estimation of a 3D map for dense RGB mapping in real-time. (**I**) We setup the online mapping with RGB stream as a two stage maximal likelihood problem, including the initialization and the on-the-fly continue learning phases, where the prior aims to learn mapping without poses, and the later to achieve online high quality reconstruction with poses. (**II**) Inspired by Chen et al. (2022), we leverage the factorized neural field to decompose the grid features into a lower-dimensional space, slimming model while ensuring its representation ability. (**III**) We leverage the kernel function (Dexheimer & Davison, 2023) to derive the depth guidance, distributing sample on surfaces with high uncertainty and achieving speedy convergence during on-the-fly stage. To maintain the function's stability, we propose a self-supervised depth training method on Signed Distance Function (SDF) and NeRF depth. Consequently, our solution enables high quality, real-time mapping for dense RGB mapping. We show that our FMapping can reconstruct a high-fidelity dense map more efficiently than existing methods with no poses provide in the initialization. During on-the-fly phase, our method achieves real-time high-fidelity mapping with a standard PyTorch implementation, with its map quality comparable to RGB-D based methods.

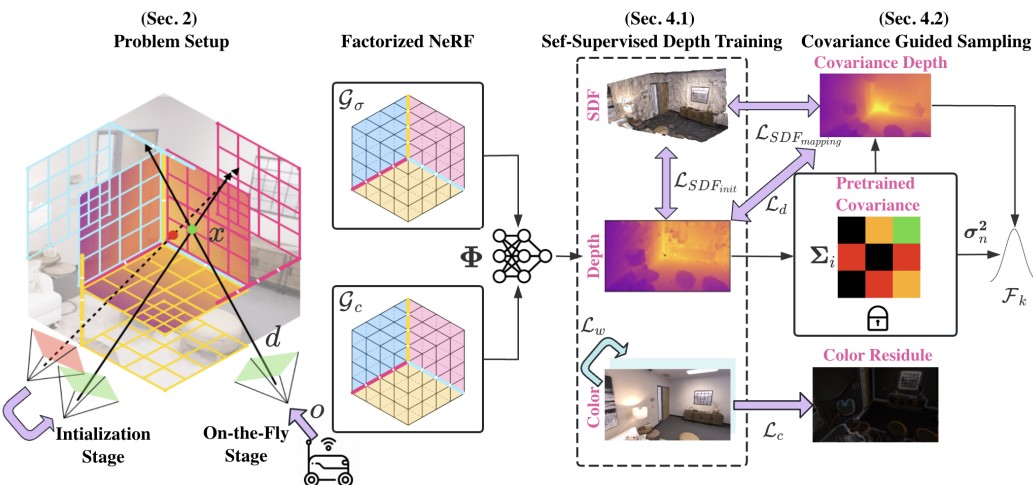

Figure 2: Schematic diagram of the framework. $G_\sigma$ and $G_c$ represent the geometric and appearance representations, respectively. By decoding representation features, the final color, depth, and SDF estimation are obtained. The entire online training process is constrained by SDF losses $L_{SDF}$ in a self-supervised way, RGB loss $L_c$, warpping loss $L_w$, and covariance depth loss $L_d$.

## 2 PROBLEM SETUP

We address the challenge of the online instant construction of dense maps with only posed RGB frames as input. In this setup, given image frames $I$ and corresponding poses $\tilde{P}$, we estimate the frames' color $\tilde{I}$ and depth $\tilde{D}$ to reconstruct dense RGB map $\tilde{m}$. We assume that poses obtained from trackers entail a normally distributed disturbance $n$, formulated as $\tilde{P} = P + n, n \sim \mathcal{N}(0, Q)$. Our goal is instant dense reconstruction upon receiving a posed RGB stream of any accumulated length, i. e. $I_{1:k}$. Inspired by Montemerlo et al. (2002), We formulate the dense mapping problem as estimating a conditional joint probability distribution of:

$$P(\tilde{m}|\tilde{P}_{1:k}, I_{1:k}). \tag{1}$$

NeRF (Mildenhall et al., 2020) has been introduced as a compelling solution for implicit scene representation. Recent works (Chan et al., 2021; Chen et al., 2022; Johari et al., 2022) leverage the matrix decomposition to speed up NeRF computation, i. e. , representing the high-dimensional features by samples' 3D coordinates along with their latent feature, which has emerged as a prevailing technique for NeRF acceleration. Denote NeRF function as $\mathcal{G}$ and its decoder as $\Phi$, the dense mapping problem of 1 is estimating the map $\tilde{m} = \Phi(\mathcal{G}(r))$ to maximize the posterior probability:

$$\tilde{m} = \arg \max_{(\Phi, \mathcal{G}, \tilde{r})} P(\tilde{P}_{(k-w):k}, I_{(k-w):k}|\Phi(\mathcal{G}(\tilde{r}_{(k-w):k}))), \tag{2}$$

where $\tilde{r}(t) = \tilde{o} + t\tilde{d}$ are samples drawn from camera rays originated from the center $\tilde{o}$ with normalized direction $\tilde{d}$, $t$ denotes the camera distance from $\tilde{o}$ to any point sample at $\tilde{r}$. In practice, only partial frame observation is cached in a window of size $w$ to achieve real-time operation.

As shown at Fig. 2, to best align our solutions to the real-world setting, e. g. , unstable trackers, we make two assumptions. **1) Noisy start** with large $Q$ dominates the system optimization during the kick-off of mapping. **2) Stable continuous learning** with mapping uncertainty $\Sigma$ takes major role and has limited noise $n$. Specifically, it can be divided into **Initialization stage** which needs to predict $\tilde{I}$, $\tilde{D}$ and $\tilde{P}$; and the **On-the-fly mapping stage** that estimates $\tilde{I}$, $\tilde{D}$ upon the system's initialization with stabilized pose stream. Therefore, we can view the implicit map construction as a maximal likelihood problem, which is equivalent to minimizing its quadratic form:

$$\arg \min_{(\Phi, \mathcal{G}, \tilde{r}, \tilde{P})} (\tilde{I}_{init} - I_{0:w})^T Q^{-1} (\tilde{I}_{init} - I_{0:w}),$$

$$\arg \min_{(\Phi, \mathcal{G}, \tilde{r})} (\tilde{I} - I_{(k-w):k})^T \Sigma^{-1} (\tilde{I} - I_{(k-w):k}). \tag{3}$$

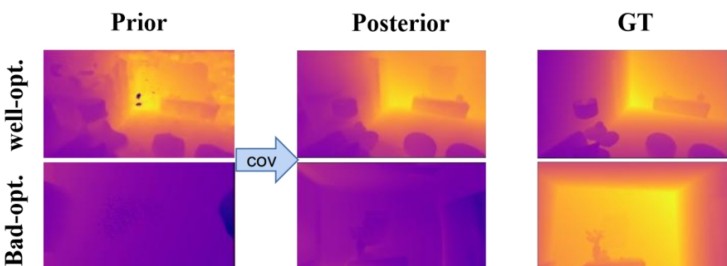

Figure 3: Visual demonstration of covariance depth estimation based on well-optimized neural implicit representation and bad-optimized.

The subsequent issue is how to design a system that constrains the two-stage uncertainty of the system under specified conditions. Given the large window size (Li et al., 2023) or provided depth (Bian et al., 2023), the neural implicit representations can be decently optimized with extremely noisy pose inputs or even no pose inputs. Specifically, during the initialization of the system, these methods use cross-frame consistent constraint to minimize the pose variance $Q$ given a relatively large window size. However, for continuous instant mapping, the large sliding window size is not an optimal choice, since it is hard to converge within limited optimization iterations. Inspired by Dexheimer & Davison (2023), we want to explicitly constrain the covariance $\Sigma$ in Eq. 3 by a cost-effective pre-trained kernel function. The idea is to forecast the correlation between the depths of any two pixels upon receiving an RGB frame. In this way, a covariance function can be constructed to model depth distribution of the current frame, i.e., we can obtain strong depth prior over RGB frames by modelling the depth function $f$ over any pixels $x$ and $x'$ as a Gaussian Process ($\mathcal{GP}$): $f(\mathbf{x}) \sim \mathcal{GP}\left(m(\mathbf{x}), k(\mathbf{x}, \mathbf{x}')\right)$. In this way, pre-trained pixel-wise covariance can be leveraged to infer kernel function $k(\mathbf{x}_i, \mathbf{x}_j)$ to provide highly reliable depth priors. Conditioning on them, we proposed a covariance-guided sampling in 4.2 to stabilize instant reconstruction.

Despite of the effectiveness of covariance functions, it highly depends on the well-estimated depth distribution of previous frames, as showed in Fig. 3. Therefore, it's crucial to maintain a relatively accurate depth to guide covariance function to infer depths without self-supervised training described in Sec. 4.1.

## 3 RELATED WORKS

**Dense Visual SLAM.** Dense visual SLAM has experienced rapid evolution in the past two decades. Compared to sparse visual SLAM algorithms (Klein & Murray, 2007; Mur-Artal & Tardós, 2017) that reconstruct sparse point clouds, dense visual SLAM algorithms (Newcombe et al., 2011b) are able to recover dense point cloud representations of the scene. Some iconic traditional dense SLAM works (Newcombe et al., 2011a; Keller et al., 2013) explicitly represent surfaces using standard volume representation. In addition, some works Dai et al. (2017b); Vespa et al. (2018) employ hierarchical volume representations, which offer increased efficiency but present challenges in implementation and parameter optimization due to their size. Recently, deep learning-based works Czarnowski et al. (2020); Li et al. (2020; 2018) have made great advances in the dense visual SLAM, bringing the benefits of both accuracy improvement and robustness enhancement. *Different from the aforementioned explicit representation methods, we focus on implicitly representing the scene given the posed RGB images, which is compact and can be extended to unobserved regions.*

**Monocular depth estimation.** With no depth inputs, it is necessary to introduce additional signals to supervise the depth estimation (Zhu et al., 2023). For example, NICER-SLAM employs the off-the-shelf depth estimation model (Eftekhar et al., 2021) to generate the depth ground truth. However, monocular depth estimation is an ill-posed problem due to its scale ambiguity (Bhoi, 2019). To compensate for it, NICER-SLAM adds a scale item to the depth loss. However, it relies on a heavy depth estimation model to ensure the quality of depth ground truth. In addition, the scale problem interferes with the consistency between frames. NICER-SLAM and DIM-SLAM introduce warping loss to enforce the geometric consistency between frames. However, the warping loss only supervises the colour information, instead of directly targeting the structural consistency. *Differently, we leverage the kernel function to derive the depth guidance, which can improve the scale consistency of depths between frames, while maintaining the lightweight and real-time advantages.*

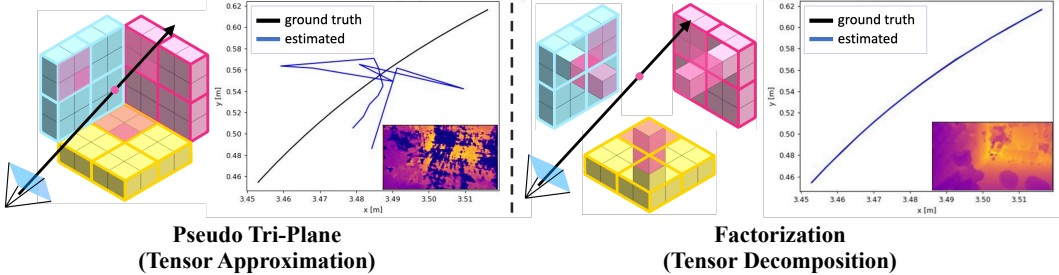

**Pseudo Tri-Plane**
**(Tensor Approximation)**

**Factorization**
**(Tensor Decomposition)**

Figure 4: Heuristic comparison of two efficient approximations of hybrid representation of Neural Radiance Filed, i. e. , Tri-projection (**Left**) and Factorization (**Right**) in the system initialization stage with only RGB supervision, which is a joint estimation of initial poses and implicit maps. The pose trajectory plot and the depth estimation suggest that the factorized 4D tensors are more robust to elusive camera trajectory and unobserved geometry.

**Implicit Dense Mapping.** Implicit representations have demonstrated their capacity to encode scenes within a latent feature space through the utilization of a single Multi-Layer Perceptron (MLP). The implicit representation has manifested in various applications across diverse domains, including novel view synthesis (Mildenhall et al., 2020; Müller et al., 2022; Verbin et al., 2022; Zhang et al., 2020), surface reconstruction (Yariv et al., 2021; Oechsle et al., 2021; Wang et al., 2021), as well as the creation and manipulation of scenes and avatars (Liu et al., 2021; Yang et al., 2021).

Neural Radiance Fields (NeRF) serve as an example that generates novel views based on sparse input data through a single MLP. It has spurred subsequent research about dense mapping. Pioneering this effort, iMAP (Sucar et al., 2021) demonstrated that a single MLP can effectively represent a 3D scene, even extending to unobserved regions. To extend the implicit dense mapping to larger scenes, NICE-SLAM (Zhu et al., 2022) employs hierarchical voxel grids and pre-trained decoders to enhance the representation capacity. Additionally, ESLAM Johari et al. (2022) replaces the voxel grids utilized in NICE-SLAM with compact feature planes, improving the speed significantly.

To reduce the demand for depth inputs, NeRF-SLAM (Rosinol et al., 2022) and Orbeez-SLAM (Chung et al., 2022) have been integrated explicitly the NeRF into visual SLAM systems. However, this integration results in redundant architectures. On the other hand, NICER-SLAM (Zhu et al., 2023) relies on heavy pre-trained geometric models, which can not meet real-time demand. Recently, DIM-SLAM (Li et al., 2023) introduced the first dense RGB SLAM system entirely based on the neural implicit mapping. However, the problem has been downgraded into estimating voxel grid occupancy using a single channel without exploiting the robustness and expressiveness of high-dimensional latent features. Additionally, such occupancy estimation requires tri-linear interpolation of stacked grids of many different resolutions, leading to an undesirable computational budget.

## 4 FMAPPING

### 4.1 SELF-SUPERVISED DEPTH TRAINING

As depicted at Fig. 2, we denote factorized neural radiance field as 4D representation $\mathcal{G}$ and propose a self-supervised strategy that leverages geometric consistency to speed up the mapping speed. The first goal is to improve efficiency. It can be achieved by decomposing NeRF feature computation in $\mathcal{G}$ by multiplying the matrix and vector in the lower dimensional space. Recent works Chan et al. (2021); Chen et al. (2022); Johari et al. (2022) leverage the matrix decomposition to speed up NeRF computation, i. e. , projecting the high-dimensional features to their low dimensional counterparts, which has emerged as a prevailing technique for NeRF acceleration. In Fig. 4, we evaluate the two common decomposition paradigms, namely the Tri-plane representation that projects the 3D tensor onto three 2D feature planes; and the BTD-based Factorized representation (Chen et al., 2022) that interprets it as multiplication between matrix and vectors. The trajectory and depth estimation results Fig. 4 show that the Factorization scheme appears to be more robust during the initialization stage. Therefore, we leverage NeRF factorization to enhance computational efficiency without dampening the rendering fidelity. To cultivate the potential of representation ability of $\mathcal{G}$ in mapping with RGB, it is a common practice to leverage geometric consistency derived from images. For instance,

Li et al. (2023) proposes to enforce geometric consistency using a warping loss $\mathcal{L}_w$ in the spirit of multi-view stereo (Zheng et al., 2014). We follow the same paradigm in Li et al. (2023) by enforcing multi-scale geometric loss:

$$\mathcal{L}_w = \frac{1}{\mathcal{M}} \sum_{\boldsymbol{q}_j \in \mathcal{M}} \sum_{j,l} \sum_{s \in \mathcal{S}} \boldsymbol{B}_{\boldsymbol{q}_j} SSIM(\mathcal{N}^s_{\boldsymbol{q}_j}, \mathcal{N}^s_{\boldsymbol{q}_{j \to l}}), j \neq l,$$

$$\boldsymbol{q}_{j \to l} = \boldsymbol{K}_l \tilde{\boldsymbol{R}_l}^T (\tilde{\boldsymbol{R}}_j \boldsymbol{K}_j^{-1} \boldsymbol{q}_j^h \tilde{D}_{\boldsymbol{q}_j} + \tilde{\boldsymbol{T}}_j - \tilde{\boldsymbol{T}}_l), \tag{4}$$

where randomly sampled $|\mathcal{M}|$ pixels and their corresponding patch $|\mathcal{N}|$ sizes $\mathcal{S}$ from cached frames inside the initialization window are re-projected to neighbouring frames. Next, Reflecting structural similarity ($SSIM$) is applied to calculate the difference inside the visibility mask $\boldsymbol{B}$. The 2D pixel $\boldsymbol{q}_j$ in frame $j$ is lifted into 3D space and then project it to another frame $l$ represented by $\boldsymbol{q}_{j \to l}$. $\tilde{\boldsymbol{P}}$ is the estimated poses, and $\boldsymbol{q}_j^h$ is the homogeneous coordinates of $\boldsymbol{q}_j$, $\boldsymbol{K}$ is the camera intrinsic.

## 4.2 COVARIANCE GUIDED SAMPLING

Given NeRF function $\mathcal{G}$ and its decoder $\Phi$, jointly denoted as parameters $\theta$, density and colors are estimated by underlying continuous volumetric scene function $\sigma_\theta(\tilde{\boldsymbol{x}})$ and $c_\theta(\tilde{\boldsymbol{x}}, \tilde{\boldsymbol{d}})$. Volume rendering (Mildenhall et al., 2020) aims to enhance spatial coherence by integrating estimated samples $\tilde{\boldsymbol{x}}$ along rays $\tilde{r}$ for color supervision,

$$\tilde{\boldsymbol{I}}(\tilde{\boldsymbol{r}}) = \sum_{i=1}^N \alpha_\theta(\tilde{\boldsymbol{x}}_i) \prod_{n<i} (1 - \alpha_\theta(\tilde{\mathbf{x}}_n)) c_\theta(\tilde{\mathbf{x}}_i, \tilde{\boldsymbol{d}}),$$

$$\alpha_\theta(\tilde{\boldsymbol{x}}_i) = 1 - exp(-\sigma_\theta(\tilde{\boldsymbol{x}}_i)\delta_i), \tag{5}$$

where $\alpha_\theta(\tilde{\boldsymbol{x}}_i)$ denotes the penetrating light at $\tilde{\boldsymbol{x}}_i$ and then composites the sample radiance into the rendered frames. Therefore, following the conventions, e. g. Zhu et al. (2022), the depth $\tilde{\boldsymbol{D}}_k$ and color $\tilde{\boldsymbol{I}}_k$ of pixel can be formulated as:

$$\tilde{\boldsymbol{D}}_k = \sum_{i=1}^N w_\theta(\tilde{\boldsymbol{x}}_i) t_i, \ \tilde{\boldsymbol{I}}_k = \sum_{i=1}^N w_\theta(\tilde{\boldsymbol{x}}_i) c_\theta(\tilde{\mathbf{x}}_i, \tilde{\boldsymbol{d}}), \tag{6}$$

where $w_\theta(\tilde{\boldsymbol{x}}_i) = \alpha_\theta(\tilde{\boldsymbol{x}}_i) \prod_{n<i}(1 - \alpha_\theta(\tilde{\mathbf{x}}_j))$. $\alpha_\theta(\tilde{\boldsymbol{x}}_i)$(Eq. 5) entails uncertainty regarding sample distribution $\delta$. Oechsle et al. (2021) uses the sigmoid activation directly on $\sigma_\theta(\tilde{\boldsymbol{x}}_i)$ to avoid the ambiguity, i. e. , $\alpha_\theta(\tilde{\boldsymbol{x}}_i)$ is replaced by $O_\theta(\tilde{\boldsymbol{x}}_i) = Sigmoid(\sigma_\theta(\tilde{\boldsymbol{x}}_i))$ in Eq. 5.

However, as shown in Fig. 2, without direct sensor depth supervision like Sucar et al. (2021); Zhu et al. (2022); Johari et al. (2022), $\tilde{\boldsymbol{D}}_k$ is indirectly constrained through color rendering loss $\mathcal{L}_c = ||\tilde{\boldsymbol{I}}_k - \boldsymbol{I}_k||_2^2$. Inspired by recent works on instant reconstruction (Johari et al., 2022; Wang et al., 2023) which leverages Signed Distance Function (SDF) to build the underlying geometry. We designed implicit mapping to infer depth based on the rendering equation while simultaneously outputting SDF Fig. 2. Since camera distances, i. e. , depth values are equal regarding SDF and inferred depth from NeRF, it allows for self-supervision to put pixel-wise depth constraint to extend further to spatial occupancy coherence:

$$\mathcal{L}_{SDF}(\boldsymbol{\omega}) = \frac{1}{\mathcal{K}} \sum_{k \in \mathcal{K}} \frac{1}{\mathcal{P}} \sum_{i \in \mathcal{P}} \left( t_i + \Phi_{\boldsymbol{g}}(\mathcal{G}(\boldsymbol{x}_i)) \cdot Tr \cdot \boldsymbol{\omega} - \tilde{\boldsymbol{D}}_k \right)^2, \tag{7}$$

where $\Phi_{\boldsymbol{g}}(\mathcal{G}(\boldsymbol{x}_i))$ gives SDF value estimation of sampler $\boldsymbol{x}_i$, based on its distance $t_i$ start from a camera origin and truncation distance $Tr$ on any ray. $\boldsymbol{\omega}$ is weights to distribute samples according to underlying uncertainty. We set $\boldsymbol{\omega} = 1$ for $\mathcal{L}_{SDF_{init}}$ loss in the initialization stage. Furthermore, we can also use $\mathcal{L}_d = ||\tilde{\boldsymbol{D}}_k - \boldsymbol{D}_{k_*}||_2^2$ to supervise depth derived from NeRF.

At the on-the-fly stage, we assume both pose variance and representation are stabilized. As discussed in Sec. 2, we can utilize the pre-trained covariance function to infer depth maps upon receiving the newest input images. In detail, condition on geometric estimations made by neural implicit functions, we gain depths prior from predicted depth distribution (Dexheimer & Davison, 2023):

$$\mathbf{f}_* \sim \mathcal{N}\left(m(\tilde{\boldsymbol{D}}_{j \to l}), K(\boldsymbol{I}_l, \boldsymbol{I}_l)\right). \tag{8}$$

Upon receiving $\boldsymbol{I}_l$, we calculate the mean $m(\cdot)$ of the re-projected depth $\tilde{\boldsymbol{D}}_{j \to l}$ from the last frame , which is available after using photometric warping from the last frames. Specifically, the posterior distribution of the depth function of frame $l$ can be calculated to obtain its predictive covariance depth $\tilde{\boldsymbol{D}}_{l_*}$ and covariance $\Sigma_{l_*}$:

$$
\begin{aligned}
\tilde{\boldsymbol{D}}_{l_*} &= m(\tilde{\boldsymbol{D}}_{j \to l}) + K_{\text{fn}}(K_{\text{nn}} + \sigma_n^2 I)^{-1}(\tilde{\boldsymbol{D}}_{j \to l} - m(\tilde{\boldsymbol{D}}_{j \to l})), \\
\Sigma_{l_*} &= K_{\text{ff}} - K_{\text{fn}}(K_{\text{mn}} + \sigma_n^2 I)^{-1} K_{\text{nf}},
\end{aligned}
\tag{9}
$$

where $K$ stands for positive semi-definite matrix provided by covariance function given $n$ samples. $\sigma_n^2$ capture per-pixel depth estimation uncertainty and thus can be cached inside an optimization window pending to supervise the sampling procedure jointly with the $\tilde{\boldsymbol{D}}_{l_*}$ for instant neural implicit mapping. Intuitively speaking, as we assumed the existence of well estimated priors, $\sigma_n^2$ is reliable to highlight complex regions to explore in the later phase, relaxing biased surface region which entails large uncertainty for better estimation. Therefore, we weights pixels in an element-wise way by $\boldsymbol{\omega} = \sigma_n^2$ in Eq. 7 for $\mathcal{L}_{SDF_{mapping}}$ loss in the on-the-fly stage.

At last, as shown in Fig. 2, we adjust the sampling distribution to consider high-reward regions around the objects' surface, i. e. , indicated by reliable covariance depth $\tilde{\boldsymbol{D}}_{l_*}$. During NeRF's rendering process, $N$ samples $X_r = \{\tilde{\boldsymbol{x}}_i | \tilde{\boldsymbol{x}}_i = \tilde{\boldsymbol{r}}(t_i), t_i < t_{i+1}\}_{i=1}^N$ along any estimated ray $\tilde{\boldsymbol{r}}$ are drawn from the coarse stratified sampling, followed by the inverse transform sampling according to the coarse-level sampling $\mathcal{F}_{cdf}(\tilde{\boldsymbol{x}})$ over its normalized PDF scores $\alpha_\theta(\tilde{\boldsymbol{x}})$,

$$X_{r,k+1} = \mathcal{F}_{pdf}^{-1}(u) \cup X_{r,k}, \ \mathcal{F}_{cdf}(\boldsymbol{x}_{i,\tilde{k}+1}) = \sum_i P(\boldsymbol{x}_{i,k}^{\tilde{}} | \alpha_\theta(\boldsymbol{x}_{i,k}^{\tilde{}}) < u), u \in \boldsymbol{U}, \tag{10}$$

where $\boldsymbol{U} \sim Unif[0,1]$, and $k$ denotes the iteration times for multi-stage estimation, e. g. , $k = 2$ in the coarse-to-fine hierarchical sampling. The resulting adjacent sample distance is $\delta_i = |\boldsymbol{x}_{i+1} - \boldsymbol{x}_i|$ from Eq. 5. $X_{r,k}$ is sorted according to their camera distances after each sampling iteration.

Specifically, as illustrated in Fig. 2, the iterative sampling process consists of a truncation sampling $\mathcal{F}_{tr}$ over the $k_{th}$ round samples with camera distances $\boldsymbol{t}_{r,k}$. Similar to the SDF loss defined in Eq. 7, $\mathcal{F}_{tr}$ aims to sample points close to $\tilde{\boldsymbol{D}}_{l_*}$ , i. e. , the surface regions for the geometric constraint. To obtain fine level of granularity, we guide $\mathcal{F}_{tr}$ with uncertainty map $\boldsymbol{\sigma}_n^2$ of each depth map $\tilde{\boldsymbol{D}}_{l_*}$. In detail, we enlarge the truncation intervals $Tr \cdot \boldsymbol{\sigma}_n^2$ for rays corresponding to pixel regions with higher uncertainty. Then we integrate samples with the previous round, and the inverse transform sampling result $\mathcal{F}_{cdf}$ from Eq. 10 for more robust sample estimation.

$$
\begin{aligned}
X_{r,k+1} &= \mathcal{F}_{cdf}^{-1}(\boldsymbol{U}) \cup \mathcal{F}_{tr}(\boldsymbol{t}_{r,k}) \cup X_{r,k}, \\
\mathcal{F}_{tr}(\boldsymbol{t}_{r,k}) &= \tilde{\boldsymbol{o}} + \boldsymbol{D}_{l_*} + \boldsymbol{t}_{r,k} \cdot Tr \cdot \boldsymbol{\sigma}_n^2.
\end{aligned}
\tag{11}
$$

## 5 EXPERIMENTS

We provide an evaluation of our FMapping on simulated dataset (Straub et al., 2019) quantitatively and qualitatively against the common real-time neural implicit scene reconstruction benchmarks, including the RGB-D method, e. g. , iMAP (Sucar et al., 2021), NICE-SLAM (Zhu et al., 2022) and H2-Mapping (Jiang et al., 2023); and RGB method like Orbeez-SLAM (Chung et al., 2022). We

also included the experimental results of the NICE-SLAM running without depth supervision that is available in Rosinol et al. (2022).

## 5.1 EXPERIMENTAL SETTING

**Covariance guided sampling:** At each inference, the multi-stage sampling with $k = 4$ is performed, where the stratified samples are collected at the first iteration for rough distribution estimation, followed by 3 iterations of covariance guided sampling $\mathcal{F}_{tr}$. During each inference, we use the updated uncertainty map to setup truncation intervals with a max value of The sampling sizes of each stage are 32, 64, 64, and 64, respectively. For each sampling iteration, 40% samples are first retrieved from $\mathcal{F}_{cdf}^{-1}$, then the remaining 60% are picked through $\mathcal{F}_{tr}$.

**The initialization phase:** We collect 15 frames for jointly estimating initial poses and local implicit maps. Once the initialization stage is finished, the first frame is added to the global keyframe set and kept fixed. The total loss during initialization phase is denoted as $\mathcal{L}_{init} = \beta_c \mathcal{L}_c + \beta_d \mathcal{L}_d + + \beta_w \mathcal{L}_{w,s \in \{1,5,11\}} + \beta_s \mathcal{L}_{SDF_{init}}$, where $\beta$ represents a weighting factor.

**The on-the-fly mapping phase:** In the process of mapping, we maintain an active window of 20 frames, with the portion of the global and local frame the same with Li et al. (2023). 20 iterations of optimization are performed to update the map for every 5 frames. The oldest 5 local frames are removed while the new 5 incoming frames are added to the window for the next map update. we also leverage the uncertainty guidance in Eq. 11 on balancing wrapping loss $\mathcal{L}_{w,s=1}$ and SDF loss $\mathcal{L}_{SDF_{mapping}}$ besides sampling procedure. The total loss during on-the-fly stage is denoted as $\mathcal{L}_{fly} = \beta_c \mathcal{L}_c + \beta_d \mathcal{L}_d + (\beta_w \mathcal{L}_{w,s=1} + \beta_s \mathcal{L}_{SDF_{mapping}}) | \mathcal{P} \otimes \boldsymbol{\sigma}_k^2$, where $\otimes$ is the element wise multiplication operation that weights each patch with normalized uncertainty map before using them for loss calculation.

**Evaluation Metrics.** We assess the precision in terms of both geometric and photometric quality. To measure geometric accuracy, we rely on the L1 depth error, which compares the estimated and ground-truth depth maps. Note that we follow the common practice to recover the metric scale by aligning the median of the estimated depth with the ground truth, as also used in Bian et al. (2023); Zhou et al. (2017). For photometric accuracy, we use the peak signal-to-noise ratio (PSNR) to analyze the similarity between the input RGB images and the rendered images.

**Implementation Details.** All experiments are conducted on a single NVIDIA RTX 3090 GPU. The factorized representation is inspired and implemented based on the TensoRF (Chen et al., 2022) and the pre-trained depth covariance function is made available by Dexheimer & Davison (2023), which has been trained on Scannet Dataset (Dai et al., 2017a). The resolution of the factorized feature grid is computed based on the pre-defined bounding box. We implement a single-resolution factorized feature grid with a dimension set to 64. To make it comparable with existing neural implicit mapping methods using a voxel grid, the resolution is roughly calculated as a voxel size of $\sim$ 8cm, given a bounding box of size 11.8m, 8.7m, and 6.8m for three coordinates (the example is given for Replica scene room 0). The feature channels are set to 16 for both density and appearance components, respectively. Both SDF and color decoders are a two-layer MLP that explains the appearance feature. Adam optimizer (Kingma & Ba, 2014) is adopted with learning rates set to 0.02 for the grid feature updating and set to 0.001 for color decoder updating, respectively. Note that some benchmark scene e. g. *office1*, is dimming and thus lacks color variance that poses difficulty in constraining SDF, so the SDF supervision is muted for better leverage the available appearance feature. A covariance depth and its corresponding uncertainty is estimated for the incoming downsampled RGB image (set to 2.5 in the Replica case) for efficient inference. Three consecutive cached sets of RGB images, the covariance depth map and the uncertainty map are sent to our Fmapping with an additional 20 global sampled overlapped frames to jointly optimize the neural implicit representation for 30 iterations.

## 5.2 RESULTS

As shown in Tab. 1, our method demonstrates generally better geometric and photometric estimation results compared to other RGB instant mapping cases and even shows comparable performance to the state-of-the-art RGB-D mapping methods (H2-mapping). Note that we report the depth output from depth covariance (cov) inferring and neural implicit rendering (rend), respectively.

Table 1: Quantitative comparison of our proposed method's mapping performance on Replica indoor scenes.

| Method | | room0 | room2 | office0 | office1 | office2 | office3 | office4 | Avg. |
|---|---|---|---|---|---|---|---|---|---|
| iMAP | Depth L1 ↓ | 5.70 | 6.94 | 6.43 | 7.41 | 14.23 | 8.68 | 6.80 | 7.64 |
| (RGB-D) | PSNR. ↑ | 5.66 | 5.64 | 7.39 | 11.89 | 8.12 | 5.62 | 5.98 | 6.95 |
| NICE-SLAM | Depth L1 ↓ | 2.53 | 2.93 | 1.51 | 0.93 | 8.41 | 10.48 | 2.43 | 4.08 |
| (RGB-D) | PSNR. ↑ | **29.90** | 19.80 | 22.44 | 25.22 | 22.79 | 22.94 | 24.72 | 24.61 |
| H2-mapping | Depth L1 ↓ | **0.34** | **0.61** | **0.33** | **0.45** | **0.53** | **0.50** | **0.40** | **0.42** |
| (RGB-D) | PSNR. ↑ | 29.24 | **27.05** | **33.72** | **33.82** | **28.91** | **29.43** | **31.17** | **30.21** |
| NICE-SLAM | Depth L1 ↓ | 11.12 | 19.03 | 11.12 | 10.24 | 16.36 | 21.33 | 14.81 | 14.18 |
| (RGB*) | PSNR. ↑ | 18.15 | 17.82 | 20.23 | 19.14 | 15.22 | **16.12** | 17.24 | 17.76 |
| Orbeez-SLAM | Depth L1 ↓ | - | - | - | - | - | - | - | 11.88 |
| (RGB) | PSNR. ↑ | - | - | - | - | - | - | - | **29.25** |
| **FMapping** (RGB) | Depth L1 ↓ (cov) | **0.21** | **0.51** | 0.16 | **0.29** | **0.40** | **0.96** | 0.32 | **0.41** |
| | Depth L1 ↓ (rend) | **0.21** | 0.60 | **0.15** | 0.30 | 0.42 | 1.00 | **0.30** | 0.43 |
| | PSNR. ↑ | **24.32** | **26.03** | **30.20** | **36.49** | **27.26** | 16.08 | **24.94** | 26.47 |

* Note that the result of the *room1* is omitted here since the initialization stage does not generate satisfactory prior to kicking off the following Gaussian process.

Table 2: Analysis of our method in comparison with existing ones in terms of mapping speed, number of parameters, and model size growth rate (parameterized by scene side-length $L$).

| Method | Mapping Speed ↓ [s] | Memory ↓ | |
|---|---|---|---|
| | | # Param. | Grow. R. |
| iMAP (RGB-D) | 0.45 | 0.22 M | - |
| NICE-SLAM (RGB-D) | **0.13** | 12.18 M | $O(L^3)$ |
| Ours-cov (RGB) | 0.19 | ∼36.00 M | - |
| Ours-rend (RGB) | 2.40 | **0.025 M** | $O(L^2)$ |

In Tab. 2, we compare our mapping speed against common real-time RGB-D neural implicit method, i. e. iMAP (Sucar et al., 2021) and NICE-SLAM (Zhu et al., 2022). Our FMapping can achieve comparable online estimation speed. Due to the lack of absolute sensor depth, an additional dedicated dynamic sampling process is required for FMapping to approach the true geometry compared to our RGB-D counterparts, therefore resulting in more processing time. Finally, regarding memory consumption, our representation is memory efficient, We ascribe it to the factorized neural field representation. Despite our covariance depth estimator based on a pre-trained covariance function entailing a large parameter size, it is a relatively cheap geometric prior with a real-time inference capability and naturally possesses the capability of cross-frame consistency for real-time reconstruction tasks, compared to other works that leverage large pre-trained monocular depth estimator (Zhu et al., 2023).

## 6 CONCLUSION

In this paper, we present FMapping, an efficient neural field mapping technique for real-time dense RGB mapping. We leverage a light and flexible geometric prior, i.e., a depth covariance function, to continuously estimate depth based on well-optimized neural implicit mapping upon receiving RGB observations. In return, this supervises the online training of NeRF. We leverage factorized neural field representation to facilitate fast convergence with efficient memory growth. We achieve state-of-the-art RGB mapping in terms of photometric and geometric accuracy, and our results are even comparable to the performance of RGB-D dense mapping.

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
