# OpenReview forum: "Factorized Neural Radiance Field with Depth Covariance Function for Dense RGB Mapping"
_ICLR.cc/2024/Conference — Submitted to ICLR 2024_

### Official Review · Reviewer_fnZK · 2023-10-23

**Soundness:** 1 poor
**Presentation:** 2 fair
**Contribution:** 2 fair
**Rating:** 3
**Confidence:** 4

**Summary:**

This paper introduces a dense RGB Neural SLAM system utilizing a TensorRF NeRF representation, referred to as "Factorized Neural Radiance Field" in the paper, and incorporates a depth covariance method for depth guidance. The authors assert that their proposed FMapping can perform real-time, high-fidelity dense map reconstruction and achieves quality comparable to RGB-D SLAM systems.

**Strengths:**

1. The motivation behind this work is well-founded. The issues addressed, such as unstable initial pose tracking due to poor initial mapping and the challenge of real-time dense mapping while striving for high-quality maps, are valid concerns.
2. The exploration of advanced NeRF structures, such as TensorRF, shows promise for more efficient optimization.
3. Leveraging depth estimation for RGB dense SLAM to expedite convergence is a reasonable approach.

**Weaknesses:**

1. The paper's evaluation results lack conviction. There is a notable absence of visualizations for the reconstructed 3D map, 2D novel view synthesis, and pose trajectories. Moreover, there is no 3D metric evaluation of scene reconstruction quality. The solitary 2D metric in Table 1 raises suspicion. The lack of per-scene evaluation results for Orbeez-SLAM and the methodology for averaging results need clarification. The discrepancy in iMAP's PSNR compared to NICE-SLAM, even inferior to the RGB version of NICE-SLAM, requires an explanation.
2. The paper should clarify the procedure used to measure speed. It appears that NICE-SLAM is already slow, but the proposed method is even slower. This conflicts with the claim of being "more efficient than other RGB-D methods." Identifying the bottleneck responsible for this discrepancy is essential.

**Questions:**

1. Please provide more evaluations according to comments on weaknesses.
2. Please explain the bottleneck of the proposed method in efficiency.

---

### Official Review · Reviewer_veoK · 2023-10-23

**Soundness:** 2 fair
**Presentation:** 2 fair
**Contribution:** 2 fair
**Rating:** 3
**Confidence:** 4

**Summary:**

The work proposes a method for fast and accurate SLAM problem using neural radiance fields and signed distance fields. It incorporates covariance estimation of the depth estimates, and build a two-stage method. The resulting map is fast and memory efficient, largely owing to the factorization approaches from recent NeRF formulations.

**Strengths:**

The paper introduces an interesting framework that incorporates depth uncertainty into better pose estimation and mapping. They claim they reach photometric and geometric accuracy comparable to RGB-D cases, but using RGB only.

**Weaknesses:**

The overall structure is not polished, and need further revisions. In high level, the abstract is too long and vague, and the method, separated into section 2 and 4, is not easy to follow. The figures miss labels or clear definitions on what is shown. Many of the variables in equations are not defined or defined later in the text. The results are limited, which is only a single page. In detail, the reference citations often times require additional paranthesis, some sentences are incomplete, and many of the procedures are hard to understand. The incompleteness of the writing is the main concen about the paper.

Minor typos:
- Page 2 "on-the-fly continue learning phases" -> on-the-fly continual learning phases
- Page 2 "with no poses provide" -> with no poses provided
- Figure 2 "Color Residule" -> Color Residual
- In Section 3, many of references may need to be put into the paranthesis, as they are not read as part of sentences. There are so many of them, so I cannot list them all here.
- Near the bottom page 4, we need a reference to NICER-SLAM and DIM-SLAM. It seems like they appear in the last paragraph of section 3 in page 5.
- When you write 'i.e. ,', please remove the tailing space.
- In page 5, it is awkward to say "speed up the speed". Please consider rephrasing it.
- An incomplete sentence in page 6: Inspired by recent works on instant reconstruction (Johari et al., 2022; Wang et al., 2023) which leverages Signed Distance Function (SDF) to build the underlying geometry.
- In page 7, third line "condition" -> "conditioned"
- In page 7, "we weights" -> "we weigh"

**Questions:**

- in Page 2, the authors claim "it would be more realistic to learn mapping from the scratch without poses from trackers". However, the current paper also uses the initial poses from trackers. Can you better claify what you are trying to mention here?

- Equation 2 is a bit strange. According to the sentence right above the eqution, $\tilde{m} = \Phi(\mathcal{G}(r))$, but the equation says that $\tilde{m} = \arg \max P(\tilde{P}, I | \Phi(\mathcal{G}(\tilde{r})))$. What is the difference between $\tilde{r}$ and $r$, and what is equation 2 conditioned on?

- Q in the last paragraph of page 3 is not defined until page 4. This should be re-structured.

- Equation 3 is hard to follow. The equation is missing all the values that it is optimizing for ($\Phi, \mathcal{G}, \tilde{r}, \tilde{P}$). What is the dimension of $I$ and $Q$? $I$ is an image, or it appears to be a temporal stack of images, which might be in a 3D tensor format. Q, on the other hand, seems to be a $w \times w$ matrix (actually Q is never clearly defined). Do you assume a single variance value per time step, which include both rotation and translation?

- The description in Figure 3 is too vague. Please be more specific.

- What is "large window size" in page 4?

- In page 4, how do you genreate the depth ground truth? Can you refer what is generated by a neural network as "ground truth"?

- It is not clear what Figure 4 is depicting. Please increase the fonts for axes, and label what is shown. What do you mean by tensor approximation and tensor decomposition? What is tri-projection and what factorization? I believe tri-plane representation can also be considered as a kind of decomposition. Please clearly define what you are refering to. Please provide detailed evidence in you conclusion about "elusive camera trajectory and unobserved geometry".

- In Section 4.1, the authors claim that "The trajectory and depth estimation results Fig. 4 show that the Factorization scheme appears to be more robust during the initialization stage." Why is it?

- I don't understand Equation 4. I think the first term in Equation 4 needs to be $1/|\mathcal{M}|$ with the absolute value. Also please define j and l. I am not sure why the visibility mask depends on j only, but not l. Also, $\tilde{P}$ needs to be replaced by $\tilde{T}$ and $\tilde{R}$ because the equation does not have $\tilde{P}$.

- $\delta_i$ in equation 5 is not defined.

- $t_i$ in equation 6 is not defined.

- What is the exact network structure for SDF? Do you use a shared architecture to output both density/color and SDF?

- In equation 7, we might need absolute value signs to indicate the number of elements for $\mathcal{K}$ and $\mathcal{P}$.

- What is truncation distance Tr?

- What is K in equation 8?

- In Equation 9, what are indices - j, l, f, n, m? Also, please avoid using 'm' as one of the indices because you are using it for a map.

- What are the differences between $\tilde{D_{j \rightarrow l}}$  and $m(\tilde{D_{j \rightarrow l}})$?

- Can you try applying factorization to other works as well, and compare the memory?

---

### Official Review · Reviewer_3ryB · 2023-10-30

**Soundness:** 1 poor
**Presentation:** 2 fair
**Contribution:** 2 fair
**Rating:** 3
**Confidence:** 5

**Summary:**

This paper proposes an RGB-based neural implicit mapping method, namely F-Mapping. The authors provide some theoretical analysis for the problem definition, and approach it in two stages: initialisation and on-the-fly mapping. The proposed method uses a factorised tensor field as the scene representation for mapping, showing better results at the initialisation stage compared with another efficient representation, i.e. Tri-plane. A pre-trained covariance depth prior is leveraged to improve mapping quality in the on-the-fly mapping stage. Some quantitative and qualitative results are shown on a synthetic dataset.

**Strengths:**

1. Using factorised tensor field as scene representation in online neural implicit mapping seems novel. Qualitative comparison also shows its advantage over alternative efficient representation.

2. The method achieves competitive mapping results compared to previous RGB-D and RGB-based system.

**Weaknesses:**

1. While the theoretical analysis of problem setup seems interesting, it doesn’t seem to provide new insights beyond existing practice in the community. The decomposition into initialisation and on-the-fly optimisation have already been used by previous methods [1].

2. Some equations in theoretical analysis are also confusing. For example: Why Eq (2) is called posterior instead of likelihood? How are $Q$ and $\Sigma$ defined? If Q represents the uncertainty of pose estimation, how can it be used to formulate the quadratic form in Eq (3)? The dimension mismatches.

3. The proposed self-supervised SDF supervision in Eq (7) is very similar to the pseudo-SDF supervision used in previous methods [2, 3, 4, 5]. The only difference seems to be that instead of using measured depth values it uses rendered depth. While this modification seems new, it’s also a bit confusing as the rendered depth value itself also comes from the predicted SDF. There should be more analysis about this formulation, or at least an ablation study is needed to empirically show it works better.

4. In the proposed covariance-guided sampling, why is $\sigma_{n}$ instead of $diag(\Sigma_{l_*})$ used to measure depth uncertainty? Is this a typo?

5. This uncertainty value is also used to dynamically modulate the truncation distance in self-supervised SDF loss. In principle, the truncation distance should be a scene-specific hyper-parameter and shouldn't change its value for each ray. What's the reason for having ray-dependant truncation values? There is also no analysis or ablation studies.

6. The sampling process also seems very complicated, combining uniform stratified, importance and covariance depth-guided, but no ablation study was provided to show it has really improved the result. In the formulation of Eq (9), the depth variance only depends on the sampled pixel location, which means it will not change when increasing sampling stages. Then why do we need to apply the guided sampling in multiple stages?

7. The experiments were not thoroughly done. It only shows comparisons to two RGB-based method, some important baselines, such as [1], [6] are missing. Besides, the proposed method uses a strong pre-trained depth prior, then it is not entirely fair to compare with NICE-SLAM in RGB mode without providing any depth priors. In addition, Ablation studies are missing for every proposed component.

[1] DIM-SLAM: Dense RGB SLAM With Neural Implicit Maps, ICLR'23

[2] Neural RGB-D Surface Reconstruction, CVPR'22

[3] GO-Surf: GO-Surf: Neural Feature Grid Optimization for Fast, High-Fidelity RGB-D Surface Reconstruction, 3DV'22

[4] ESLAM: Efficient Dense SLAM System Based on Hybrid Representation of Signed Distance Fields, CVPR'23

[5] Co-SLAM: Joint Coordinate and Sparse Parametric Encodings for Neural Real-Time SLAM, CVPR'23

[6] DROID-SLAM: Deep Visual SLAM for Monocular, Stereo, and RGB-D Cameras, NeurIPS'21

**Questions:**

Please see weakness.

---

### Official Review · Reviewer_WxZo · 2023-11-02

**Soundness:** 1 poor
**Presentation:** 1 poor
**Contribution:** 2 fair
**Rating:** 3
**Confidence:** 3

**Summary:**

The paper deals with the task of real-time SLAM using a single RGB camera. From a sequence of images of a stationary scene, the goal is to compute the associated camera poses and depth maps. The authors argue that although many neural mapping and SLAM systems have been proposed, none of them are suitable for real-time applications. For computational efficiency, the authors have used existing ideas in the literature – (a) the idea of factorized radiance fields from the TensorRF paper, and (b) the idea of a learned depth covariance function proposed by Dexheimer and Davison 2023. They describe the implementation of a two-staged pipeline. If I understand correctly, then the first stage of the pipeline computes camera poses and predicts initial noisy depth maps and the second stage (on-the-fly stage) refines the depth maps.
The on-the-fly stage uses a Gaussian process for estimating a refined depth map and the NeRF-based neural nets that have been trained during the first stage.
The method is evaluated on the Replica dataset and the authors claim that their method is on par with existing RGB-D methods which assume that depth from a depth sensor is available as input.

**Strengths:**

The main idea of the paper is to speed-up the RGB-based mapping process by using key ideas published recently – (1) A factorized radiance field representation that was proposed in by Chen et al 2022 to make radiance field learning more efficient, and (2) leveraging the concept of learning a depth covariance function and using it for monocular depth estimation. The individual ideas are not novel but perhaps they have not been combined earlier in the way it is proposed here.

**Weaknesses:**

My main concern is that the technical exposition is currently quite poor. The draft needs to be thoroughly revised for better readability. I have listed some typos, comments, suggestions, below but it is not an exhaustive list and the poor readability made it difficult for me to judge the merits of the main idea proposed in the paper. The paper is not ready for publication.

The claim that the proposed method is for online mapping, and suitable for real-time mapping, is incorrect. As described in the paper, there are two stages. It is not clearly stated, but to the best of my understanding, the first stage is where all the image frames are processed and several quantities are learned or estimated, – the neural networks representing the radiance fields, the camera poses and the depth covariance function. Then in the second stage, the images are revisited and depth estimates are refined by leveraging the radiance field learned in the first stage, as well as the camera poses and the depth covariance function, etc. (based on the technical overview in Section 2 and the graphics presented in Figure 2). The second stage quite efficient and has low running time (5 Hz). However, it is incorrect to call such a system an online SLAM or mapping system as the whole system including the stage that computes poses is not running in real-time.

After reading Section 4.1 and 4.2 carefully, I was unable to understand how the camera poses are computed either during the initialization stage or afterwards, i.e. whether they are further refined during the on-the-fly stage. Various losses are defined in the various equations, but it is not completely clear how they are used in the algorithm.

The evaluation is based on the L1 depth error between the estimation and GT depth map, but to compute the relative scale, the median depth values in the estimated and GT depth maps are used. This metric is not appropriate as it does not penalize scale drift in the reconstruction. Instead, a more appropriate approach is to compute a single relative scale factor for the complete scene reconstruction, and using that in the metrics computation.

-------- comments, typos, etc. -------------

Section 4.1:
“we denote factorized neural radiance field as 4D representation G … “ --> Earlier in the paper, the term G is used to denote the NeRF function when describing prior work, but the mathematical form was not precisely defined. It is unclear whether the function G is the same as one that was used in previous papers, or different. Nor is the factorization of G actually explained anywhere. Considering that “Factorized Neural Radiance Field” is in the paper title and one of the core ideas behind the proposed approach, it must be properly defined and explained.

Typos:

page
“scarify their real-time ...” -> sacrifice their real-time ..,? not sure.
“we tackle to” -> incorrect grammar
“on-the-fly continue “ -> continuous,

“where the prior aims to learn mapping without poses” – what does this phrase mean? Please rephrase and explain clearly.

“later to” should be “latter to”

“To maintain the function’s stability”  does function refer to the kernel function? The term “stability” needs a proper definition and explanation.

During on-the-fly phase, our method achieves real-time high-fidelity mapping  when does the initialization phase end and the on-the-fly phase start?

page3:
“warpping loss”
“We formulate .. “  we formulate …

Section 2: Problem Setup
“… In this setup, given image frames I and corresponding pose \tilde{P}, we estimate the frames’ color \tilde{I} and depth \tilde{D} to reconstruct dense RGB map \tilde{m}.
But it is unclear what \tilde{m} actually is? It needs to be defined. This makes it difficult to understand what Equation (2) actually means. I can try to guess, but my concern is that the reader should not have to guess these things. Similarly, I have a hard time understanding what the functions G and \Phi (decoder) take as input and return as output.

page 4:
Despite of -> Despite
The following sentence is unclear:
"Therefore, it's crucial to maintain a relatively accurate depth to guide covariance function to infer depths without self-supervised training described in Sec 4.1." -- the sentence is confusing because it talks about self-supervised training and inference at the same time. Also, it is unclear how the goal to "maintain accurate depth" is actually achieved.

page 5:
“To cultivate the potential of representation ability of .. “  -- sentence is unclear. Must be rephrased.

page 6:
“… calculate the difference inside the visibility mask B.” -- what is B? it needs to be defined. Where does B come from, i.e. how is it computed?
“denotes the penetrating light”  this phrase should be rephrased.

**Questions:**

- Are camera poses parameterized as SE(3), how are they optimized?

- Regarding the statement on page2, "Different from the aforementioned explicit representation methods, we focus on implicitly representing the scene given the posed RGB images, which is compact and can be extended to unobserved regions", it implies that explicit methods are not compact or cannot be extended to unobserved regions, but this is not true. Please clarify.

- Regarding the statement on page2, "Differently, we leverage the kernel function to derive the depth guidance, which can improve the scale consistency of depths between frames, while maintaining the lightweight and real-time advantages." -- in monocular RGB reconstruction, there is no way to recover the absolute scale. I suppose you are referring to relative scale in that statement. Then, how is the relative scale consistency guaranteed throughout the whole sequence?

---

### Meta-Review · Area_Chair_h27D · 2023-12-04

**Metareview:**

A single RGB input for the scene reconstruction with NeRF is proposed in this paper. Experiments demonstrate the effectiveness of the proposed method.

The paper received four “reject” ratings. All the reviewers gave “poor” or “fair” for the soundness, presentation, and contribution. The main problems are as follows:
1)	The writing needs great improvement.
2)	The evaluation metric is not appropriate.
3)	New insights are needed.
4)	The theoretical analysis is confusing.
5)	More analyses or ablation studies are needed.
6)	The experiments were not thoroughly done.
7)	The paper's evaluation results lack conviction.
8)	The paper should clarify the procedure used to measure speed.

The authors didn’t give any responses or discussions for the reviews.

Based on the above comments, the decision was to reject the paper.

**Justification For Why Not Higher Score:**

The paper received four “reject” ratings. All the reviewers gave “poor” or “fair” for the soundness, presentation, and contribution. The main problems are as follows:
1)	The writing needs great improvement.
2)	The evaluation metric is not appropriate.
3)	New insights are needed.
4)	The theoretical analysis is confusing.
5)	More analyses or ablation studies are needed.
6)	The experiments were not thoroughly done.
7)	The paper's evaluation results lack conviction.
8)	The paper should clarify the procedure used to measure speed.

The authors didn’t give any responses or discussions for the reviews.

**Justification For Why Not Lower Score:**

N/A

---

### Decision · Program_Chairs · 2024-01-16

Reject